# Promotional Effects of Rare-Earth Praseodymium (Pr) Modification over MCM-41 for Methyl Mercaptan Catalytic Decomposition

**Xiaohua Cao** [1,2,3], **Jichang Lu** [1,2,3,*], **Yutong Zhao** [1,2,3], **Rui Tian** [1,2,3], **Wenjun Zhang** [1,2,3], **Dedong He** [2,3,4] and **Yongming Luo** [1,2,3,*]

1   Faculty of Environmental Science and Engineering, Kunming University of Science and Technology, Kunming 650500, China; caoxiaohua72@126.com (X.C.); zhaoyutong816@126.com (Y.Z.); Tianruia2021@126.com (R.T.); z15198861570@126.com (W.Z.)
2   The Innovation Team for Volatile Organic Compounds Pollutants Control and Resource Utilization of Yunnan Province, Kunming 650500, China; dedong.he@kust.edu.cn
3   The Higher Educational Key Laboratory for Odorous Volatile Organic Compounds Pollutants Control of Yunnan Province, Kunming 650500, China
4   Faculty of Chemical Engineering, Kunming University of Science and Technology, Kunming 650500, China
*   Correspondence: lujichangc7@kust.edu.cn (J.L.); environcatalysis@kust.edu.cn (Y.L.)

**Abstract:** Praseodymium (Pr)-promoted MCM-41 catalyst was investigated for the catalytic decomposition of methyl mercaptan ($CH_3SH$). Various characterization techniques, such as X-ray diffraction (XRD), $N_2$ adsorption–desorption, temperature-programmed desorption of ammonia ($NH_3$-TPD) and carbon dioxide ($CO_2$-TPD), hydrogen temperature-programmed reduction ($H_2$-TPR), and X-ray photoelectron spectrometer (XPS), were carried out to analyze the physicochemical properties of material. XPS characterization results showed that praseodymium was presented on the modified catalyst in the form of praseodymium oxide species, which can react with coke deposit to prolong the catalytic stability until 120 h. Meanwhile, the strong acid sites were proved to be the main active center over the 10% Pr/MCM-41 catalyst by $NH_3$-TPD results during the catalytic elimination of methyl mercaptan. The possible reaction mechanism was proposed by analyzing the product distribution results. The final products were mainly small-molecule products, such as methane ($CH_4$) and hydrogen sulfide ($H_2S$). Dimethyl sulfide ($CH_3SCH_3$) was a reaction intermediate during the reaction. Therefore, this work contributes to the understanding of the reaction process of catalytic decomposition methyl mercaptan and the design of anti-carbon deposition catalysts.

**Keywords:** rare earth; MCM-41; $CH_3SH$ decomposition; stability; reaction mechanism

## 1. Introduction

Methyl mercaptan ($CH_3SH$) is one of the typical unconventional volatile organic compound (VOC) species, which exhibits high toxicity, volatility, and causticity [1,2]. People exposed to high concentrations of $CH_3SH$ for a long time contributes to the presentation of mental illnesses [2,3]. In addition, methyl mercaptan not only participates in a series of complex reactions in air pollution events, such as PM2.5 (Fine Particulate Matters, $\varphi < 2.5\mu m$) [4], ozone layer depletion, and acid rain [4,5], but also corrodes equipment and poisons the catalyst during various processes. More importantly, there are wide industrial sources for the $CH_3SH$ emission, including wood-pulping industries, waste water treatment, sanitary landfills, petroleum refining, the natural gas industry and energy-related activities [6–9]. Therefore, on account of increasingly rigorous catalytic technical standards and stringent environmental regulations, improved removal of $CH_3SH$ is imperative.

The reported methods currently used for the removal of $CH_3SH$ include alkaline treatment [10], biological degradation [11], adsorption [12–14] incorrect ref order, 12 detected after 9. You jumped the numbers in between., hydrodesulfurization, catalytic

oxidation [15,16], and catalytic decomposition [17–21]. Among the above, the method of catalytic decomposition has been considered an effective option because of its low waste, lack of need of other reactants, and its ease of implementation. At present, nano-$CeO_2$ and HZSM-5-based catalysts have attracted wide attention for the process of the catalytic decomposition of $CH_3SH$ owing to their high concentration of oxygen vacancies [18] and tunable acid–base properties [19], respectively. However, these catalysts have drawbacks in the form of poor stability and the high temperatures required for complete conversion. Huguet et al. [19] showed that poisonous $CH_3SH$ can be converted into $CH_4$ and $H_2S$ small-molecule products at high temperature (600 °C) using pure HZSM-5 zeolite. However, the stability for $CH_3SH$ abatement over HZSM-5 catalyst was no more than 10 h. Furthermore, Lu et al. [22] reported that the rare earth (RE) elements La-modified HZSM-5 catalyst could increase basic sites (lanthanum oxy-carbonates composites) and decrease strong acid sites to improve the stability (nearly 80 h time-on-stream (TOS) test). However, the deactivation of above HZSM-5-based catalysts were due to the blockage of microporous channels by deposited coke. Compared with HZSM-5-based catalysts, the conversion of $CH_3SH$ can reach 100% at low temperature (450 °C) over nano-$CeO_2$ [20,23] and modified $Ce_{0.75}RE_{0.25}O_{2-\delta}$ (RE = Sm, Gd, Nd, and La) [17,24], but the ultrashort stability was still presented for these catalysts (about 8 h TOS test) without obvious improvement. These results suggest that the problem of deposited coke over these two types of catalysts for $CH_3SH$ decomposition has not been resolved yet, due to the fact that the reaction mechanism of the catalytic degradation of methyl mercaptan is ambiguous. As reported in the literature, praseodymium plays a vital role in removing coke during the methane reforming reaction [25,26]. Praseodymium, as an adjacent element with cerium, can exhibit good stability due to the presence of various stoichiometrically defined oxides ($Pr_nO_{2n-1}$), like cerium oxide, which can accelerate the elimination of coke deposit [25–28]. Sierra et al. [25] added Pr into $La_{1-x}A_xNiO_{3d}$ perovskites as catalyst, which shows the highest catalytic activity and stability in the dry reforming of methane via the reaction between $PrO_2$ and carbon residues gasifying the carbon deposits [26]. Abello et al. [29] indicated that a series of Pr (0–7 wt.%)-modified $MgAl_2O_4$ spinel oxide-supported Ni catalysts decreased the deactivation rate by influencing the type and amount of coke deposit. Therefore, praseodymium has been deemed a good promoter to eliminate carbon deposits for methyl mercaptan abatement.

Moreover, it has been evidenced that the catalytic performance is significantly affected by different supports. Coke deposition ultimately leads to catalyst rapid deactivation by blocking the microporous channel of HZSM-5 support [19,22,30]. $Al_2O_3$ carrier has a high content of strong acidic sites, which results in easily deposited coke on the surface of catalyst compared with HZSM-5 [31–33]. Among numerous support materials, mesoporous MCM-41 support has been regarded as a potential support candidate because of its large specific area, small diffusion hindrance, and uniform-sized pores [34–40]. As far as we know, no investigation on Pr-supported MCM-41 catalyst for catalytic decomposition of $CH_3SH$ has been conducted so far.

On the basis of this background, a certain amount of Pr was impregnated to the MCM-41 support with large specific area and small diffusion hindrance, and the obtained Pr-modified MCM-41 sample was used for methyl mercaptan elimination. The synthesized samples were carried out by $N_2$ adsorption–desorption, X-ray diffraction (XRD), $NH_3$-TPD, $CO_2$-TPD, $H_2$-TPR, and X-ray photoelectron spectrometer (XPS) characterizations to investigate the effect of the modification of rare metal Pr on the physicochemical properties of the MCM-41 zeolite catalyst. In addition, the reaction mechanism of eliminating methyl mercaptan was investigated in detail by product distribution results, which were detected by flame ionization detector (FID) and flame photometric detector (FPD).

## 2. Experimental

### 2.1. Chemicals and Materials

Praseodymium nitrate hexahydrate, cetyltrimethylammonium bromide (CTAB), tetraethylorthosilicate (TEOS), ammonia, and other chemical reagents were obtained from Aladdin Biochemical Technology Co., Ltd, Shanghai, China. All of the above chemicals were at least analytical grade. Methyl mercaptan (5000 ppm $CH_3SH$, $N_2$ as balance gas) was purchased from Dalian Special Gases Co. Ltd., Dalian, China.

### 2.2. Catalyst Synthesis

#### 2.2.1. The Preparation of the MCM-41 Support

Organic–inorganic hybrid mesoporous silica materials were prepared by the sol-gel method. Cetyltrimethylammonium bromide (CTAB) and tetraethylorthosilicate (TEOS) were used as template and silicon source, respectively. An ammonium solution ($NH_3 \cdot H_2O$) was applied as pH control agent. Firstly, 14 g of CTAB and 54 mL of $NH_3 \cdot H_2O$ were dissolved at room temperature into 635 mL of deionized water. Subsequently, 57.6 mL of TEOS was slowly added into the mixture under continuous stirring. The obtained gel was filtered and washed with deionized water, dried at 105 °C for 24 h, and then calcined at 550 °C for 5 h to remove the template.

#### 2.2.2. The Synthesis of Pr/MCM-41 Catalyst

The Pr/MCM-41 sample was successfully synthesized with an incipient wetness impregnation method using praseodymium nitrate hexahydrate ($Pr(NO_3)_3 \cdot 6H_2O$, AR, 99.95%) as the precursor and mesoporous MCM-41 molecular sieve as support. Firstly, the calculated amount of $Pr(NO_3)_3 \cdot 6H_2O$ was dissolved in 8 mL of deionized water to obtain 10 wt% Pr. Then 2 g of the MCM-41 support was mixed with the above precursor solution with continuous stirring. Subsequently, the resulting mixture was placed in an oven and dried at 110 °C for 12 h. After that, the dried sample was calcined at 550 °C for 5 h. The obtained sample was denoted as 10% Pr/MCM-41 catalyst.

### 2.3. Characterization

Nitrogen adsorption–desorption isotherms were conducted on a NOVA 4200e Surface Area and Pore Size Analyzer with the assistance of the NovaWin-CFR software. The samples were degassed under high vacuum at 300 °C for 3 h before analysis and then measured at −196 °C. The Brunauer, Emmett & Teller Method (BET) was used to calculate the specific surface areas of catalyst. The pore size distribution and pore volume were obtained from the Barrett, Joyner & Halenda Method (BJH). X-ray diffraction (XRD) patterns were obtained with a Rigaku D/max-1200 diffractometer equipped with Cu K$\alpha$ radiation in the range of 10° to 90° at a scanning rate of 10°/min. The surface acidity and basicity properties of catalysts were respectively characterized by the temperature-programmed desorption of ammonia ($NH_3$-TPD) and carbon dioxide ($CO_2$-TPD), using gas chromatography (GC) with a thermal conductivity detector (TCD). In the $NH_3$-TPD experiments, all catalysts (100 mg) were first treated under helium (He, 30 mL/min) at 450 °C for 30 min and cooled down to 100 °C. Then, 30 mL/min of $NH_3$ (10 vol.% $NH_3$ in He) flow was introduced into the reactor for 1 h to saturate the acid sites of the measured samples. In order to remove the physiosorbed ammonia, the prepared sample was swept with He for 1 h. Finally, a $NH_3$-TPD analysis was carried out at a gas flow rate of 30 mL/min in He, ranging from 100 to 800 °C (10 °C/min ramp rate). Meanwhile, $CO_2$-TPD was analyzed with similar conditions to the $NH_3$-TPD procedure. Additionally, the adsorption $CO_2$ at 30 °C and the desorption was registered from 30 °C to 900 °C. The redox properties of the sample were generally performed by hydrogen temperature-programmed reduction ($H_2$-TPR) in a fixed-bed with a quartz tube reactor. Before $H_2$-TPR measurements, 100 mg of the catalyst was pretreated in the flow of mixture gas (10 vol.% $H_2$/Ar, 30 mL/min) at 100 °C for 1 h to remove adsorbed water. The heating rate of $H_2$-TPR results, ranging from 100 °C to 800 °C, was 10 °C/min in the 10 vol.% $H_2$/Ar. A TCD detector recorded the $H_2$ consumption

during the measurement. X-ray photoelectron spectroscopy (XPS) was measured on an Escalab 250Xi spectrometer by using Al Kα anode as the excitation source. The binding energies (BE) of Pr 3d, O 1s, and Si 2p were corrected by the C 1s peak at 284.8 eV.

### 2.4. Catalytic Activity and Stability Measurement

The catalytic decomposition $CH_3SH$ experiment was carried out in a fixed bed microreactor under atmospheric pressure. First, 0.2 g of catalyst was sieved to sizes in the range of 40 to 60 mesh and placed into a quartz tube reactor. $CH_3SH$ as the reactive gas, with the concentration of 5000 ppm (30 mL/min), was introduced into the system. Then, 0.2 g of the 10% Pr/MCM-41 catalyst was used to evaluate the catalytic stability at 600 °C. The reactants were detected by the online gas chromatography (GC) equipped with FID and FPD detectors. The catalytic conversion of $CH_3SH$ ($X_{CH3SH}$) was defined as follows:

$$X_{CH_3SH}(\%) = \frac{C_{[CH_3SH]_{in}} - C_{[CH_3SH]_{out}}}{C_{[CH_3SH]_{in}}} \times 100\%$$

Here, $C_{[CH3SH]in}$ and $C_{[CH3SH]out}$ represent the initial concentration of methyl mercaptan before the reaction and the remaining concentration after the reaction, respectively. The reaction rate of $CH_3SH$ was calculated as μmol of $CH_3SH$ converted per second per $S_{BET}$ as follows:

$$\text{Reaction rate} = \frac{\left(0.5\% \times \frac{V}{V_m}\right) \times X_{CH_3SH}(\%)}{S_{BET} \times m_{(catalyst)}}$$

where 0.5% is the concentration of $CH_3SH$, V is the gas flow rate of $CH_3SH$, and $V_m$ is the molar volume at one standard atmosphere. $S_{BET}$ is determined based on the BET method. $m_{(catalyst)}$ represents the quality of catalyst during the activity test.

## 3. Results and Discussion

### 3.1. Structure of Support and Modified Pr/MCM-41 Catalyst

Nitrogen adsorption–desorption isotherms and pore size distribution of fresh MCM-41 and 10% Pr/MCM-41 catalysts are displayed in Figure 1A,B, and the related data are shown in Table 1. From Figure 1A, the isotherms of MCM-41 and 10% Pr/MCM-41 catalysts are related to a IV-type with a H1 type hysteresis loop [41–43], which reflects the typical mesoporous structure. The mesoporous structure of the modified Pr/MCM-41 catalyst did not notably change. From Figure 1B, after the praseodymium species was introduced, the intensity of the pore diameter centered at about 3.7 nm significantly decreased and some new pore appeared at 5 nm, indicating some praseodymium species was deposited on the external surface, and the others were introduced in the porosity, filling the pores of the MCM-41 the support. Table 1 lists the specific surface areas and pore volume of these two catalysts. The $S_{BET}$ and pore volume of the unmodified MCM-41 support were measured to be 1289 m$^2$/g and 0.379 cc/cm$^3$, respectively. After the addition of 10 wt% Pr loading, the $S_{BET}$ and pore volume of the 10% Pr/MCM-41 catalyst significantly decreased to 920 m$^2$/g and 0.111 cc/cm$^3$, respectively, which was a normal phenomenon when some additional species was introduced into the supports.

**Table 1.** Textural properties of fresh MCM-41 and 10% Pr/MCM-41 catalysts.

| Sample | $S_{BET}$ (m$^2$/g) | Pore Volume (cc/cm$^3$) | Pore Diameter (nm) |
|---|---|---|---|
| MCM-41 | 1289 | 0.379 | 3.742 |
| 10% Pr/MCM-41 | 920 | 0.111 | 3.382 |

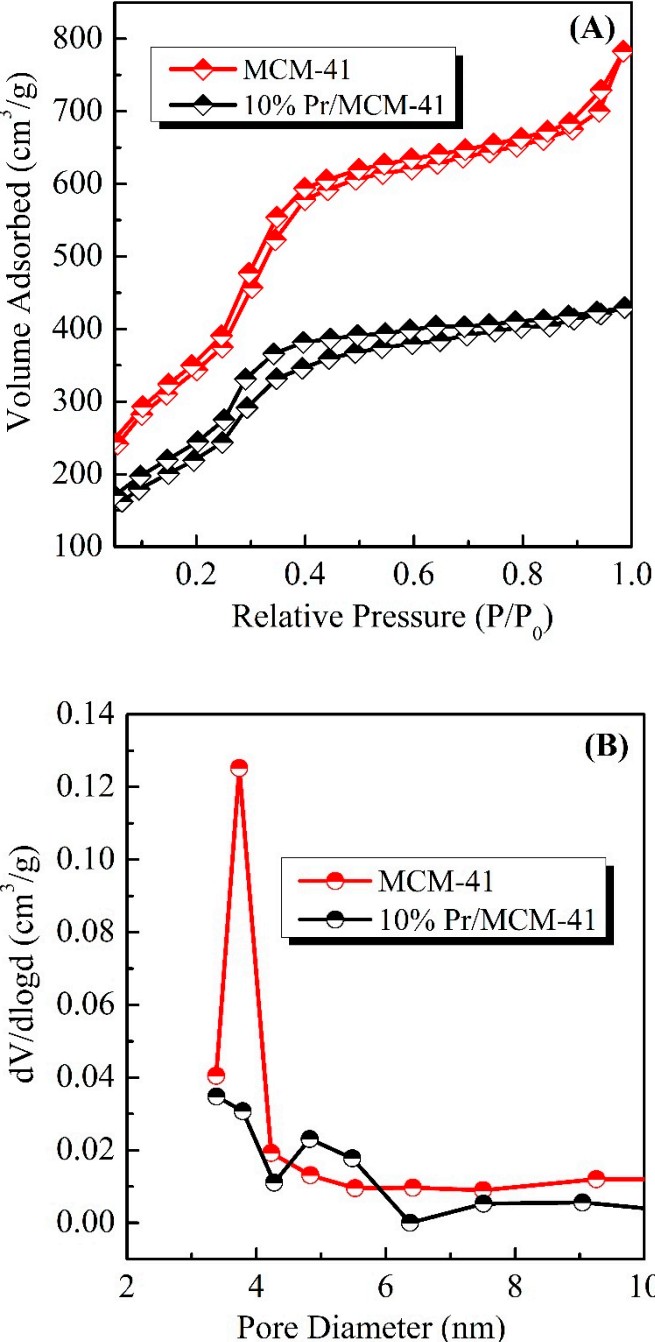

**Figure 1.** The $N_2$ adsorption–desorption isotherms (**A**) and pore size distribution plot (**B**) of the fresh MCM-41 and 10% Pr/MCM-41 catalysts.

The structural property of catalysts was characterized by X-ray diffraction (XRD), as shown in Figure 2. The pattern consisted of a very broad peak for 10% Pr/MCM-41. Compared with the MCM-41 support, the peak intensities weakened obviously, indicating that the hexagonal mesoporous structure of MCM-41 was partly damaged after the addition of Pr. This may have been due to some Pr species entering and blocking the pores of the MCM-41 support [43]. According to the standard powder diffraction files of $Pr_6O_{11}$, no corresponding pattern of any bulk praseodymium species phase was observed for the supported catalysts, indicating that praseodymium species were well dispersed on the MCM-41 support [44].

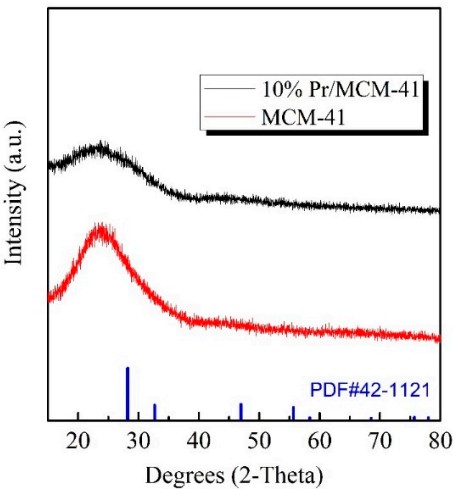

**Figure 2.** The detailed X-ray diffraction (XRD) profile of the fresh MCM-41 and 10% Pr/MCM-41 catalysts compared with the standard powder diffraction files of $Pr_6O_{11}$.

### 3.2. The Surface Acid–Base and Reducibility of Support and Pr/MCM-41 Catalysts

The surface acidity and basicity were performed via the $NH_3$-TPD and $CO_2$-TPD measurements to evaluate the 10% Pr/MCM-41 catalyst (Figure 3). As shown in Figure 3A, a small $CO_2$ desorption peak, observed at 100 °C to 300 °C in the low-temperature region, is assigned to the adsorption of surface OH groups, which can be associated with the weak basic sites. Meanwhile, the high-temperature desorption peak at 600 °C to 850 °C, connected with coordinatively unsaturated $O_2$ anions, is ascribed to the strong basic sites [45–47]. These basic sites are conducive to the adsorption of acidic methyl mercaptan molecules. Figure 3B shows the $NH_3$-TPD profiles of the 10% Pr/MCM-41. The high-temperature desorption peak at ~700 °C is ascribed to the strong acid sites [48–50], which can cleave the C-S bond to remove $CH_3SH$ [23,31]. As is well known, the pure MCM-41 support did not have obvious basic and acid sites [43,51–53]. After the modification of Pr, the strength of the acid-base sites is significantly enhanced, which is beneficial to the adsorption and activation of acid $CH_3SH$ molecules. Figure 4 shows $H_2$-TPR profiles of the fresh MCM-41 and 10%Pr/MCM-41 samples, which were used to investigate the redox properties of catalysts. However, no obvious peaks were detected over the catalyst due to the highly dispersed $Pr_6O_{11}$ species over the mesoporous MCM-41 support [28].

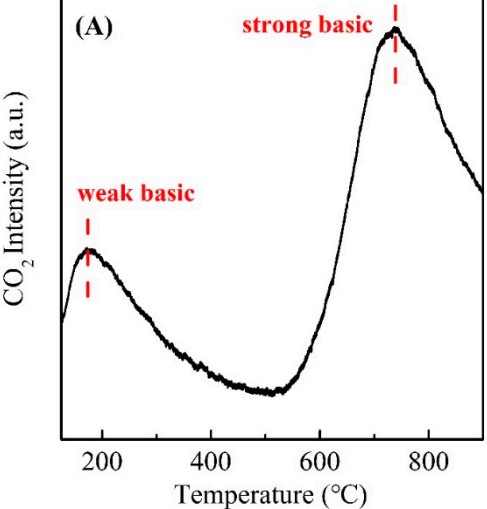

**Figure 3.** *Cont.*

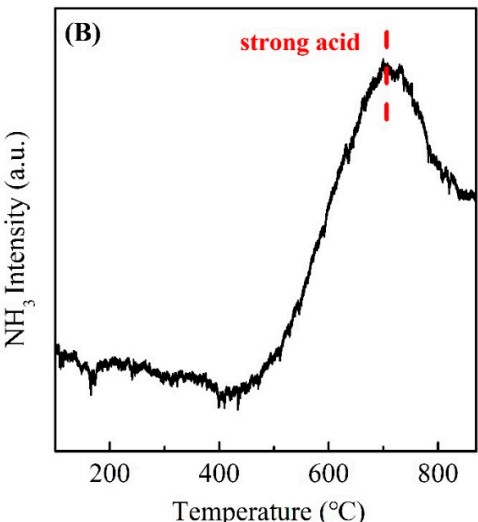

**Figure 3.** $CO_2$-TPD (**A**) and $NH_3$-TPD (**B**) patterns of the 10% Pr/MCM-41 catalyst.

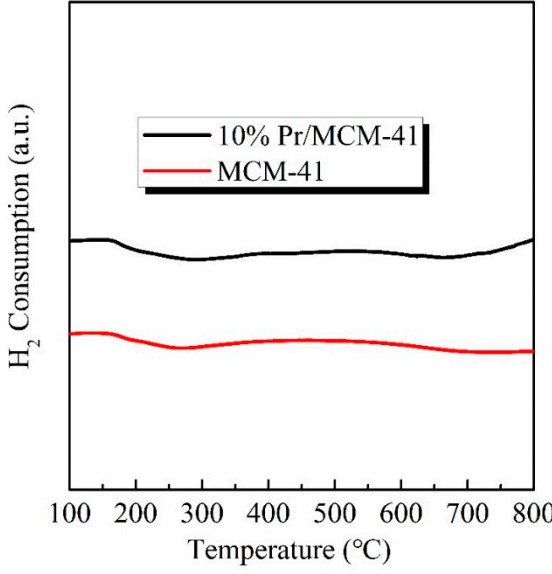

**Figure 4.** $H_2$-TPR profiles of MCM-41 and 10% Pr/MCM-41 catalysts.

*3.3. The Surface Composition of the Support and Pr/MCM-41 Catalysts*

XPS spectra were conducted to analyze the element oxidation states and chemical component of the surface species. The XPS spectra results of Pr 3d, O 1s, and Si 2p are displayed in Figure 5A–C, respectively. The binding energy position, content, and ration for each assigned peak are quantified and presented in Table 2. From Figure 5A, the peaks at 929.5 eV and 950.3 eV are attributed to the Pr 3d signal of $Pr^{3+}$ species. The peaks at 933.9 eV and 954.8 eV are attributed to $Pr^{4+}$ species, indicating the coexistence of $Pr^{4+}$ and $Pr^{3+}$ in the $Pr_6O_{11}$ species over the fresh 10% Pr/MCM-41 catalyst [50,54,55]. The ratio of $Pr^{4+}/Pr^{3+}$ of the 10% Pr/MCM-41 catalyst is 2.36, evidencing that the content of $Pr^{4+}$ is higher than that of $Pr^{3+}$ in the fresh catalyst. The O 1s spectra of the unmodified MCM-41 were only deconvoluted into a high intensity peak at 533.0 eV, which was ascribed to the oxygen species in the MCM-41 support (Figure 5B). As displayed in Figure 5C, an intense peak of MCM-41 at 103.8 eV, presented in the Si 2p XPS spectra, can be assigned to the Si-O-Si from the $SiO_2$ phase [31]. Furthermore, the O 1s and Si 2p spectra of the 10% Pr/MCM-41 were also tested for comparison. The positions of the O and Si species

both shift to the lower binding energy after the addition of Pr, indicating that there is an interaction between praseodymium and silica.

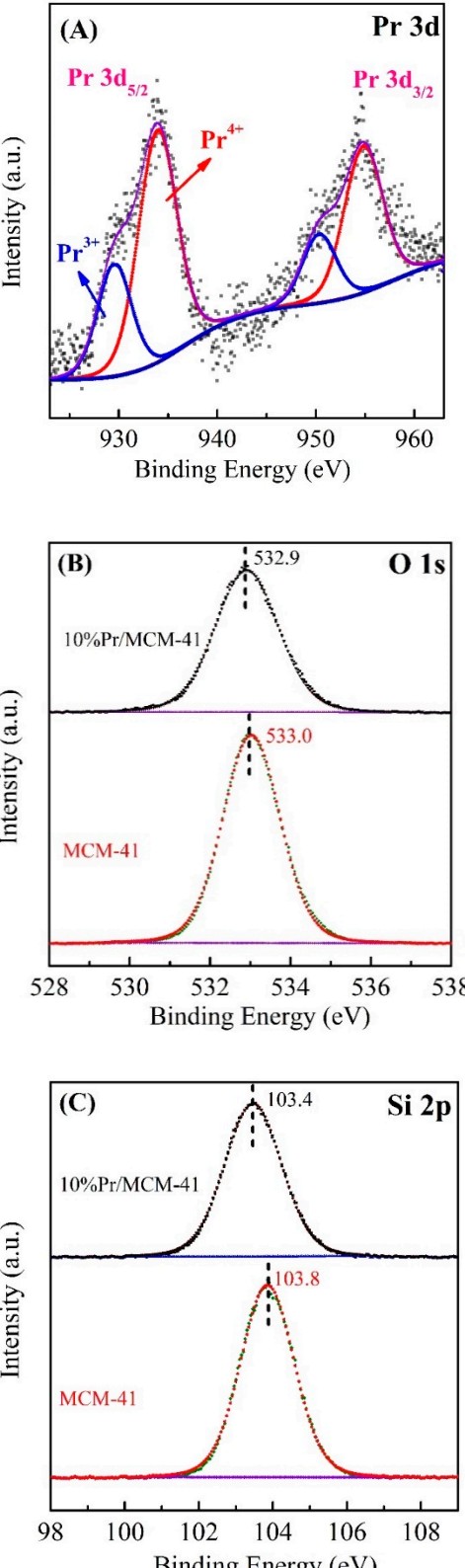

**Figure 5.** Pr 3d (**A**), O 1s (**B**), and Si 2p (**C**) X-ray photoelectron spectrometer (XPS) spectra of MCM-41 and 10% Pr/MCM-41 catalysts.

**Table 2.** Physical parameters and chemical and surface compositions of Pr, O, and Si in fresh MCM-41 and 10% Pr/MCM-41 catalysts.

| Sample | $Pr^{4+}$ | | $Pr^{3+}$ | | $Pr^{4+}/Pr^{3+}$ | Si | $O_{Si\text{-}O\text{-}Si}$ |
|---|---|---|---|---|---|---|---|
| | Content (%) | Binding Energy (eV) | Content (%) | Binding Energy (eV) | | Binding Energy (eV) | Binding Energy (eV) |
| MCM-41 | — | — | — | — | — | 103.85 | 533.03 |
| 10%Pr/MCM-41 | 70.24 | 954.83 | 29.76 | 933.95 | 2.36 | 103.47 | 532.91 |

### 3.4. CH₃SH Reaction Rate and Stability of Catalyst

The evaluation of the reaction rate and stability for $CH_3SH$ decomposition is summarized in Figure 6A,B, respectively. The reaction rates of $CH_3SH$ over MCM-41 and 10% Pr/MCM-41 catalysts are displayed in Figure 6A. It can be found that the reaction rate increases with an increase in the temperature over these two catalysts. The reaction rate of MCM-41 is 0.0001 and 0.0009 ($\mu mol \cdot s^{-1} \cdot m^{-2}$) at 400 °C and 450 °C, respectively. However, the reaction rate of the 10% Pr/MCM-41 is 0.0054 and 0.0199 ($\mu mol \cdot s^{-1} \cdot m^{-2}$) at 400 °C and 450 °C, respectively. Thus, the reaction rate of the 10% Pr/MCM-41 is nearly 54 and 22 times that of MCM-41 catalyst at 400 °C and 450 °C, indicating that MCM-41 has almost no activity in this temperature range. At 600 °C, almost a complete $CH_3SH$ conversion was achieved over the 10% Pr/MCM-41 catalyst. From Figure 6B, the variation of $CH_3SH$ conversion as the reaction time progressed on the 10% Pr/MCM-41 catalyst at 600 °C was investigated. Excellent stability can be observed in that the 10% Pr/MCM-41 maintains 100% of $CH_3SH$ conversion even after 120 h time-on-stream test. The reason why the catalyst has an ultralong stability is because the praseodymium oxide species can react with carbon residues to gasify the carbon deposits on the surface of the 10% Pr/MCM-41 catalyst.

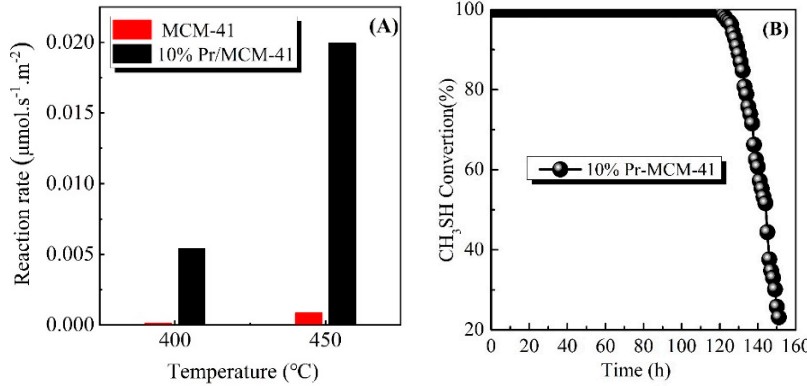

**Figure 6.** (**A**) Reaction rate of $CH_3SH$ at 400 °C and 450 °C over MCM-41 and 10% Pr/MCM-41 catalysts. (**B**) Conversion of $CH_3SH$ as a function of time at 600 °C on spent 10% Pr/MCM-41 catalyst.

### 3.5. Product Distribution

Reaction products distribution results of the 10% Pr/MCM-41 catalyst are shown in Figure 7. Apparently, the catalytic performance for eliminating $CH_3SH$ is influenced by the reaction temperature from 300 °C to 600 °C. Below 475 °C, $CH_3SH$ firstly decomposed and formed a small amount of dimethyl sulfide ($CH_3SCH_3$, DMS) and hydrogen ($H_2$), and the reaction equation was $2CH_3SH \rightarrow CH_3SCH_3 + H_2$. Then, the concentration of $CH_3SCH_3$ gradually decreased with the rising temperature, indicating that there was an intermediate species during the reaction. From 475 °C to 600 °C, $CH_3SCH_3$ was further decomposed into small molecular products such as methane ($CH_4$) and hydrogen sulfide ($H_2S$), and the reaction equation is $CH_3SCH_3 \rightarrow CH_4 + H_2S$. At the high temperature of 600 °C, a certain concentration (~1000 ppm) of COS was detected by FPD. Meanwhile, a small amount of

other sulfur and carbon species, such as $CS_2$, $C_2H_4$ and $C_2H_6$, were detected in the reaction (detailed results are not shown in Figure 7).

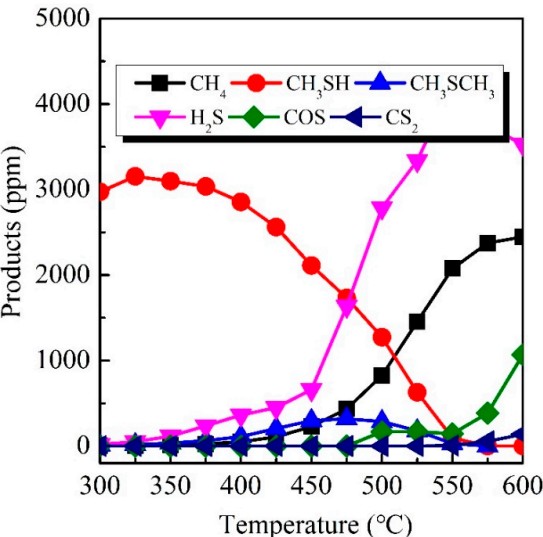

**Figure 7.** The product distribution in the 10% Pr/MCM-41 catalyst for methyl mercaptan abatement.

### 3.6. Possible Reaction Mechanism and Active Sites for the Conversion of $CH_3SH$ over the 10% Pr/MCM-41 Catalyst

It has been reported that redox and strong acid sites play an important role for eliminating $CH_3SH$. He et al. [20,23] observed that microwave-assisted rapid synthesized nano-$CeO_2$ can completely decompose $CH_3SH$ at low temperature (450 °C) due to the relatively higher redox from the abundant active oxygen species. This indicates that strong redox properties can improve the catalytic activity at low temperature. The 10% Pr/MCM-41 catalyst completely decomposes $CH_3SH$ at high temperature (600 °C) due to the poor redox. Thus, the main active sites of the 10% Pr/MCM-41 catalyst can only be strong acid sites, which can cleave the C-S bond to improve the catalytic activity.

The possible reaction mechanism for eliminating $CH_3SH$ is proposed as follows, as shown in Figure 8. At first, the acid $CH_3SH$ molecule is absorbed by the basic sites of the surface of the 10% Pr/MCM-41 catalyst. Then the absorbed reactants migrate to the active centers, such as strong acid sites. After that, the S-H bond of $CH_3SH$ firstly dissociates to $CH_3S$ and H with the 10%Pr/MCM-41 catalyst ($CH_3SCH_3$, $H_2$). Subsequently, $CH_3S$ can decompose to $CH_3$ and S via the breakage of the C-S bound by the strong acid sites. The formation of $CH_3$ and S can be hydrogenated to $CH_4$ and $H_2S$. Finally, the gas products desorb easily from the surface of the catalysts.

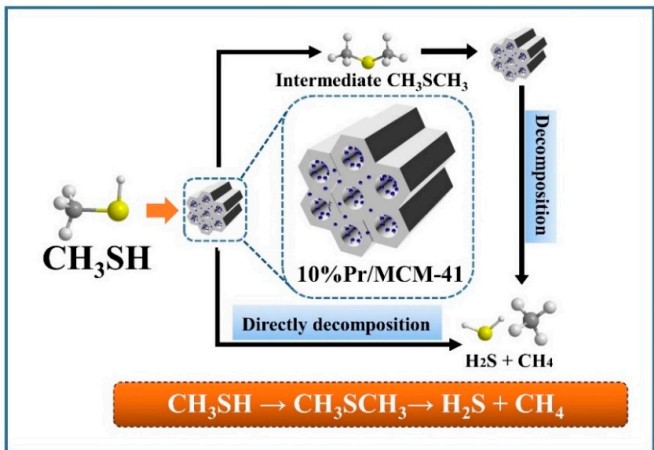

**Figure 8.** Schematic reaction mechanism for catalytically eliminating methyl mercaptan over 10% Pr/MCM-41 catalyst.

## 4. Conclusions

In this work, a Pr modified MCM-41 sample presented longer-term stability and maintained 100% of the $CH_3SH$ conversion for 120 h. The effect of the 10% Pr/MCM-41 catalyst on $CH_3SH$ catalytic decomposition was studied by a series of characterization techniques. As proved by XRD and $N_2$ adsorption–desorption results, the 10% Pr/MCM-41 sample was successfully synthesized and praseodymium oxide species were highly dispersed over the MCM-41 support. $H_2$-TPR profiles hardly showed any obvious reduction peak, indicating that the 10% Pr/MCM-41 catalyst possesses poor redox performance. Meanwhile, $CO_2$-TPD and $NH_3$-TPD results suggested that 10%Pr/MCM-41 catalyst with strong basic/acid sites promoted the adsorption and the break of C-S bonds of acid methyl mercaptan molecules. Thus, the main active sites of the 10% Pr/MCM-41 catalyst for $CH_3SH$ catalytic decomposition were the strong acid sites. In addition, the products distribution was employed to clarify the reaction mechanism. In consideration of the poor catalytic activity of the 10% Pr/MCM-41 catalyst at low temperature, the regulation of the redox performance of the catalyst in the elimination of sulfur-containing volatile organic compounds (VOCs) is the focus in the future research.

**Author Contributions:** Conceptualization: J.L. and Y.L.; Data curation: X.C., J.L. and Y.L.; Formal analysis: X.C., W.Z. and D.H.; Investigation: X.C. and Y.Z.; Methodology: Y.Z. and R.T.; Software: X.C. and R.T.; Supervision: J.L., W.Z., D.H. and Y.L.; Validation: R.T. and W.Z.; Visualization: W.Z. and D.H.; Writing–original draft: X.C.; Writing—review and editing: J.L. and Y.L. All authors have read and agreed to the published version of the manuscript.

**Funding:** This research work was supported by the National Natural Science Foundation of China (Grant Nos. 21966018, 42030712, 21667016, and 21666013), and Science Research Fund Project of Yunnan Provincial Department of Education (Grant No. 2020J0060).

**Institutional Review Board Statement:** Not applicable.

**Informed Consent Statement:** Not applicable.

**Data Availability Statement:** All data used to support the findings of this study are included within the article.

**Conflicts of Interest:** The authors declare no conflict of interest.

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
