# Peer review of "Promotional Effects of Rare-Earth Praseodymium (Pr) Modification over MCM-41 for Methyl Mercaptan Catalytic Decomposition"

_processes, doi:10.3390/pr9020400_

Round 1
Reviewer 1 Report
This draft reported the synthesis, characterization and evaluation of the Pr-modified MCM-41 zeolite catalysts for the catalytic decomposition of CH3SH. With the aid of a series materials characterization techniques, the authors studied the specific surface area, porosity, redox property, surface acidity of the materials. The authors concluded that the strong acid sites are the major active sites for the reactions, and they also demonstrated the stability and proposed a possible reaction route for the reaction. Overall, this is a comprehensive work, which deserves publication. However, I also found some defects in terms of presentation as well as some insufficient discussion on the experimental results. Therefore, I suggest a major revision before accepting this draft for publication. My comments are as follows:
- Page 1, author list, there is a mistake in the end with “and ”;
- Page 1, Abstract, the past tense should be used for the description of the experiments, please refer to the conclusion part.
- The overall English language expression should be refined to avoid grammar error and confusion, for example, Page 1 Line 39, “the industrial sources of CH3SH are very widely” should be “there are wide industrial sources for the CH2SH emission...”; Page 2 Line 52, “completely conversion” should be “complete conversion”. Other mistakes should also be corrected, especially for the Abstract, Introduction and Conclusion.
- The subtitles for Section 3. Results and discussion are not accurate. For example, Section 3.1 is titled with “Structure of support and modified Pr/MCM-41 catalyst”, however, the XRD data are presented in Section 3.2, “Textural properties of Pr/MCM-41 catalysts”, and the surface acidity and basicity are actually not the textural properties. Please reorganize the subtitles and contents for Section 3.
- Page 5, Fig. 1, the poresize distribution plot of MCM-41 and Pr/MCM-41 should also be presented along the isothermals
- Page 6, Fig. 2, the XRD profiles of both MCM-41 and Pr/MCM-41 samples should be presented in the figure and indexed properly with the PDF cards.
- Page 6, Line 211, more discussion on the activity of different Pr species should be provided according to the deconvoluted XPS profiles. The XPS data should be quantified and presented in a Table with the peak positions, ration between different surface species. The O1s spectrum should also be analyzed, discussed and presented in the table.
- Page 6, Fig. 5, make the experimental data in scatter form, and present the deconvoluted peaks of different species and background in different colors with proper legends.
- Page 10, Fig 6, the axis title for Fig 6(B) is missing. The experimental data of the MCM-41 should also be presented along the data of Pr/MCM-41 in Fig 6(B).
- Page 11, Line 263, the reaction mechanism should be presented in a schematic chart to show the process.
Author Response
Response to Reviewer 1 Comments
We thank the reviewers for their helpful and constructive comments to improve our manuscript. We have carefully made revisions as suggested. Please see our point-to-point responses below.
Point 1: Page 1, author list, there is a mistake in the end with “and ”;
Response 1: Thanks for your reminding. The corresponding mistake in author list has been revised. (Page 1 Line 4-5).
Point 2: Page 1, Abstract, the past tense should be used for the description of the experiments, please refer to the conclusion part.
Response 2: Thanks for your remarks. The tense of the abstract has been changed to the past tense with reference to the conclusion part. For example, “is, are, show” has been replaced to “was, were, showed”. Details please see Page 1 Line 16, 20, 21, 23, 26 and 27.
Point 3: The overall English language expression should be refined to avoid grammar error and confusion, for example, Page 1 Line 39, “the industrial sources of CH3SH are very widely” should be “there are wide industrial sources for the CH2SH emission...”; Page 2 Line 52, “completely conversion” should be “complete conversion”. Other mistakes should also be corrected, especially for the Abstract, Introduction and Conclusion.
Response 3: According to your suggestion, Page 1 Line 39, “the industrial sources of CH3SH are very widely” has been revised as “there are wide industrial sources for the CH3SH emission...”, see Page 1 Line 39 in the revised manuscript; Page 2 Line 52, “completely conversion” has been revised as “complete conversion”, see Page 2 Line 52 in the revised manuscript.
The grammar in the manuscript has been checked and revised by several experienced writers. We believe that the language is now acceptable for publication. Details please see our revised manuscript.
Point 4: The subtitles for Section 3. Results and discussion are not accurate. For example, Section 3.1 is titled with “Structure of support and modified Pr/MCM-41 catalyst”, however, the XRD data are presented in Section 3.2, “Textural properties of Pr/MCM-41 catalysts”, and the surface acidity and basicity are actually not the textural properties. Please reorganize the subtitles and contents for Section 3.
Response 4: Thanks for your remarks. The subtitles for Section 3 have been reorganized as follows. As you proposed, the XRD data should be classified as structure of catalyst. Thus, XRD data has been reorganized to section “3.1 Structure of support and modified Pr/MCM-41 catalyst”. The subtitle for Section 3.2 has been renamed to “The surface acid-base and reducibility of catalysts”, which used to summarize the data of NH3-TPD, CO2-TPD and H2-TPR characterization. And the subtitle for Section 3.3 has been revised as “The surface composition of catalysts”.
Details please see Page 6-8 Line 200 and 222 in our revised manuscript.
Point 5: Page 5, Fig. 1, the pore size distribution plot of MCM-41 and Pr/MCM-41 should also be presented along the isothermals
Figure 1. The N2 adsorption-desorption isotherms (A) and pore size distribution plot (B) of the fresh MCM-41 and 10% Pr/MCM-41 catalysts.
Response 5: According to your suggestions, the pore size distribution plot of MCM-41 and Pr/MCM-41 has been presented along the isothermals, as shown in Figure 1(B). From Figure 1(B), after the praseodymium species was introduced, the intensity of pore diameter centered at about 3.7 nm significantly decrease and some new pore appeared at 5 nm, indicating some praseodymium species is deposited on the external surface, and the others was introduced in the porosity, filling the pores of the mesoporous MCM-41 support. It is a normal phenomenon when some additional species was introduced into the supports.
This result has been added into the revised manuscript (please see Page 4 Line 168-182 and the revised Figure 1(B)).
Point 6: Page 6, Fig. 2, the XRD profiles of both MCM-41 and Pr/MCM-41 samples should be presented in the figure and indexed properly with the PDF cards.
Figure 2. The detailed XRD profile of the fresh MCM-41 and 10% Pr/MCM-41 catalyst compared with the standard powder diffraction files of Pr6O11.
Response 6: According to your suggestions, the XRD profiles of MCM-41 and Pr/MCM-41 samples has been presented in the Figure 2. As we known, the most common oxide of Pr is Pr6O11 (PDF cards: 42-1121). No corresponding pattern of any bulk Pr6O11 species phase is observed for the supported catalysts, indicating that praseodymium species are well dispersed on MCM-41 support. Compared with MCM-41 support, the peak intensity of modified Pr/MCM-41 catalyst obviously decreased, indicating some praseodymium species were covered on the surface of MCM-41 support.
This result has been added into the revised manuscript (please see Page 6 Line 189-196 and the revised Figure 2).
Point 7: Page 6, Line 211, more discussion on the activity of different Pr species should be provided according to the deconvoluted XPS profiles. The XPS data should be quantified and presented in a Table with the peak positions, ration between different surface species. The O1s spectrum should also be analyzed, discussed and presented in the table.
Figure 5. Pr 3d (A), O 1s (B) and Si 2p (C) XPS spectra of MCM-41 and 10% Pr/MCM-41 catalysts.
Table 2. Physical parameters, chemical and surface compositions of Pr, O and Si in fresh MCM-41 and 10% Pr/MCM-41 catalysts.
sample |
Pr4+ |
Pr3+ |
Pr4+/ Pr3+ |
Si |
O Si-O-Si |
||
Content (%) |
Binding Energy (eV) |
Content (%) |
Binding Energy (eV) |
Binding Energy (eV) |
Binding Energy (eV) |
||
MCM-41 |
— |
— |
— |
— |
— |
103.85 |
533.03 |
10%Pr/MCM-41 |
70.24 |
954.83 |
29.76 |
933.95 |
2.36 |
103.47 |
532.91 |
Response 7: Thanks for your remarks. The XPS spectra of Pr 3d, O 1s and Si 2p are displayed in Figure 5(A), 5(B) and 5(C), respectively. The binding energy position, content and ratio for each assigned peak are quantified and presented in Table 2. These two high intensity peaks of MCM-41 support at O 1s and Si 2p XPS spectra were assigned to the oxygen species and Si from the SiO2 phase, respectively. The peak intensity of Si 2p peak obviously decreased and the binding energy shifted lower binding energy after the addition of Pr, indicating that there is an interaction between praseodymium and silica.
This result has been added into the revised manuscript (please see Page 8 Line 223-237 and the revised Figure 5.
Point 8: Page 6, Fig. 5, make the experimental data in scatter form, and present the deconvoluted peaks of different species and background in different colors with proper legends.
Figure 5. Pr 3d (A), O 1s (B) and Si 2p (C) XPS spectra of MCM-41 and 10% Pr/MCM-41 catalysts.
Response 8: According to your suggestions, the XPS spectra of Pr 3d, O 1s and Si 2p have been made in scatter form. The deconvoluted peaks of both different species and background have been presented in different colors and with proper legends. Details please see Page 9-10 Line 238-241 and the revised Figure 5.
Point 9: Page 10, Fig 6, the axis title for Fig 6(B) is missing. The experimental data of the MCM-41 should also be presented along the data of Pr/MCM-41 in Fig 6(B).
Figure 6. (A) Reaction rate of CH3SH at 400 ℃ and 450 ℃ over MCM-41 and 10% Pr/MCM-41 catalysts; (B) Conversion of CH3SH as a function of time at 600 ℃ on spent 10% Pr/MCM-41 catalyst.
Response 9: Thanks for your reminding. The axis title for Fig 6(B) has been added, as shown Figure 6(B). The stability experimental data of the MCM-41 are not provided along the data of Pr/MCM-41 in Figure 6(B). This is mainly due to the following reasons: 1) Since the reaction rates of the MCM-41 and 10% Pr/MCM-41 catalysts are great disparity (Figure 6(A)), we speculate that it is difficult to completely convert methyl mercaptan on the MCM-41 catalyst at 600 ℃. However, methyl mercaptan is a highly reactive volatile sulfur compound, which exhibits high levels of toxicity to human health in view of its low odor threshold. The 13th Five-Year Plan for the Prevention and Control of VOCs pollution in China stipulated explicitly that sulfur-containing VOCs, represented by methyl mercaptan (CH3SH), were listed as one of the three major controlled pollutants and the corresponding concentration should not exceed 0.002 mg/m3. If the stability test is carried out on MCM-41 catalyst for a long time and the methyl mercaptan is not completely converted. That will put the laboratory in an extremely dangerous situation. Thus, the stability experimental of the MCM-41 cannot be carried out. 2) According to the editor’s suggestions, the reviewers' comments and the revised file should be uploaded within 8 days. Fresh MCM-41 support needs to be re-synthesized and stability experiment also needs to be tested. It is very difficult to complete the above work within 8 days, especially, considering the repeated rebound of COVID-19 epidemic situation.
Point 10: Page 11, Line 263, the reaction mechanism should be presented in a schematic chart to show the process.
Figure 8. Schematic reaction mechanism for catalytically eliminate methyl mercaptan over 10% Pr/MCM-41 catalyst.
Response 10: According to your suggestions, the reaction mechanism for catalytically eliminate methyl mercaptan over 10% Pr/MCM-41 catalyst has been presented in our revised manuscript. Please see Page 13 Line 297.

Reviewer 2 Report
This contribution has potential importance to publish in this journal.
The results and conclusions are clear; however the reviewer is wondering about the effect of the metal loading on the structural change and activities.
The authors did not notice significant change on its structure by metal loading, but by looking at the XRD, there's obviously significant damage on hexagonal structure of the MCM-41, because the peak intensities are too low. The reviewer requires the authors' comment on it.
Also, if possible, the authors have better try and show the results using the other sample of the catalysts with different Pr amount.
Author Response
Response to Reviewer 2 Comments
We thank the reviewers for their helpful and constructive comments to improve our manuscript. We have carefully made revisions as suggested. Please see our point-to-point responses below.
Point 1: The results and conclusions are clear; however, the reviewer is wondering about the effect of the metal loading on the structural change and activities.
Also, if possible, the authors have better try and show the results using the other sample of the catalysts with different Pr amount.
Response 1: Thanks for your remarks. According to the editor’s suggestions, the reviewers' comments and the revised file should be uploaded within 8 days. Fresh MCM-41 support needs to be re-synthesized and MCM-41 supported with different Pr loading also needs to be synthesized and tested. It is very difficult to complete the above work within 8 days, especially, considering the repeated rebound of COVID-19 epidemic situation.
Point 2: The authors did not notice significant change on its structure by metal loading, but by looking at the XRD, there's obviously significant damage on hexagonal structure of the MCM-41, because the peak intensities are too low. The reviewer requires the authors' comment on it.
Figure 2. The detailed XRD profile of the fresh MCM-41 and 10% Pr/MCM-41 catalyst compared with the standard powder diffraction files of Pr6O11.
Figure 1. The N2 adsorption-desorption isotherms (A) and pore size distribution plot (B) of the fresh MCM-41 and 10% Pr/MCM-41 catalysts.
Response 2: According to your suggestion, the XRD data of MCM-41 and the PDF cards of Pr6O11 have been presented in Figure 2. Compared with MCM-41 support, the peak intensity of modified Pr/MCM-41 catalyst obviously decreased, indicating the hexagonal structure of the MCM-41 is partly damaged after the addition of Pr. In addition, no corresponding pattern of any bulk praseodymium species phase is observed for the supported catalysts, indicating that praseodymium species are well dispersed on MCM-41 support.
Moreover, the pore size distribution plot of MCM-41 and Pr/MCM-41 has been presented along the isothermals, as shown in Figure 1(B). From Figure 1(B), after the praseodymium species was introduced, the intensity of pore diameter centered at about 3.7 nm significantly decrease and some new pore appeared at 5 nm, indicating some praseodymium species is deposited on the external surface, and the others was introduced in the porosity, filling the pores of the mesoporous MCM-41 support. It is a normal phenomenon when some additional species was introduced into the supports.
This result has been added into the revised manuscript (please see Page 4 Line 173-177, Page 6 Line 189-196 and the revised Figure 1,2).

Reviewer 3 Report
Authors describe a simple method for synthesizing Praseodymium (Pr) modified mesoporous silica (MCM-41) for catalytic decomposition of methyl mercaptan. They characterized the Pr promoted MCM-41 catalyst with X-ray diffraction, N2 adsorption-desorption, temperature programmed desorption of NH3 and CO2, hydrogen temperature programmed reduction, and X-ray photoelectron spectroscopy.
Authors identified Praseodymium oxide on the catalyst surface and speculate that it plays a role to react with coke deposit on electrode surface, which prolongs the catalytic activity to 120 h, which is superior to HZSM-5 zeolite.
Authors speculate on the role of strong acid sites of catalyst and propose a simple reaction mechanism of catalytic decomposition of methyl mercaptan through dimethyl sulfide intermediate.
I recommend thorough revision of the whole manuscript for typos (Line 142 “hating” --> heating; Line 157 “C[41]in” --> C[CH3SH]in; etc.). Abbreviations should be defined at first occurrence.
Dimensions should be indicated in formulae (Lines 156 and 161).
My main concern is that Authors failed to show control measurements with unmodified MCM-41 (Figs. 2, 3, 4, 5, 6). I can recommend considering the paper for publication only after addressing this concern.
Author Response
Response to Reviewer 3 Comments
We thank the reviewers for their helpful and constructive comments to improve our manuscript. We have carefully made revisions as suggested. Please see our point-to-point responses below.
Point 1: I recommend thorough revision of the whole manuscript for typos (Line 142 “hating” --> heating; Line 157 “C[41]in” --> C[CH3SH]in; etc.). Abbreviations should be defined at first occurrence.
Response 1: According to your suggestion, the manuscript has been checked and revised by several experienced writers. We believe that the language is now acceptable for publication. For example, Line 58 “TOS” --> “time-on-stream (TOS)”; Line 93 “FID and FPD” --> “flame ionization detector (FID) and flame photometric detector (FPD)”; Line 143 “hating” --> heating; Line 159 “C [41]in” --> C[CH3SH]in; etc. Details please see our revised manuscript.
Point 2: Dimensions should be indicated in formulae (Lines 156 and 161).
Response 2: Thanks for your reminding. Dimensions have been indicated in formulae. Details please see Page 4 Lines 157 and 162.
Point 3: My main concern is that Authors failed to show control measurements with unmodified MCM-41 (Figs. 2, 3, 4, 5, 6). I can recommend considering the paper for publication only after addressing this concern.
Figure 2. The detailed XRD profile of the fresh MCM-41 and 10% Pr/MCM-41 catalyst compared with the standard powder diffraction files of Pr6O11.
Figure 4. H2-TPR profiles of MCM-41 and 10% Pr/MCM-41 catalysts.
Figure 5. Pr 3d (A), O 1s (B) and Si 2p (C) XPS spectra of MCM-41 and 10% Pr/MCM-41 catalysts.
Figure 6. (A) Reaction rate of CH3SH at 400 ℃ and 450 ℃ over MCM-41 and 10% Pr/MCM-41 catalysts; (B) Conversion of CH3SH as a function of time at 600 ℃ on spent 10% Pr/MCM-41 catalyst.
Response 3: Thanks for your remarks. According to your suggestion, the control measurements data of unmodified MCM-41 have been presented in Figure 2, 4, 5 and 6. The above analysis of experimental data have been added into the revised manuscript.
NH3-TPD and CO2-TPD results of unmodified MCM-41 are not performed due to the following two reasons:1) As is well known, the pure MCM-41 did not have obvious acid and basic sites (Chem. Eng. J. 2016, 294, 343-352; J. Porous Mater. 2016, 23, 1329-1338; Chem. Eng. Technol. 2018, 41, 2186-2195; Microporous Mesoporous Mater. 2020, 297). Thus, the acidity and basicity of the modified catalyst can be compared with the results reported in the literature; 2) According to the editor’s message, the reviewers' comments and the revised file should be upload within 8 days. But this work is almost impossible to complete in eight days. The test procedure of NH3-TPD and CO2-TPD characterization is very complicated and time-consuming, especially, considering the repeated rebound of COVID-19 epidemic situation.
The stability experimental data of the MCM-41 cannot be presented along the data of Pr/MCM-41 in Figure 6(B). This is mainly due to the following two reasons:1) Since the reaction rates of the MCM-41 and 10% Pr/MCM-41 catalysts are great disparity (Figure 6(A)). We speculate that it is difficult to completely convert methyl mercaptan on the MCM-41 catalyst. However, methyl mercaptan is a highly reactive volatile sulfur compound, which exhibits high levels of toxicity to human health in view of its low odor threshold. The 13th Five-Year Plan for the Prevention and Control of VOCs pollution in China stipulated explicitly that sulfur-containing VOCs, represented by methyl mercaptan (CH3SH), were listed as one of the three major controlled pollutants and the corresponding concentration should not exceed 0.002 mg/m3. If the stability test of MCM-41 catalyst is carried out for a long time and the methyl mercaptan is not completely converted, the laboratory will be located in an extremely dangerous situation. Considering the safety issue, the stability experiment of MCM-41 cannot be carried out; 2) According to the editor’s message, the reviewers' comments and the revised file should be upload within 8 days. Fresh MCM-41 support needs to be re-synthesized and stability experiment also needs to be tested. It is very difficult to complete the above work within 8 days.

Round 2
Reviewer 1 Report
My questions and comments have been well addressed, I suggest publication of this draft as it is.